# Entropy Controllable Direct Preference Optimization

## Abstract

In the post-training of large language models (LLMs), Reinforcement Learning from Human Feedback (RLHF) is an effective approach to achieve generation aligned with human preferences. Direct Preference Optimization (DPO) allows for policy training with a simple binary cross-entropy loss without a reward model. The objective of DPO is regularized by reverse KL divergence that encourages mode-seeking fitting to the reference policy. Nonetheless, we indicate that minimizing reverse KL divergence could fail to capture a mode of the reference distribution, which may hurt the policy's performance. Based on this observation, we propose a simple modification to DPO, H-DPO, which allows for control over the entropy of the resulting policy, enhancing the distribution's sharpness and thereby enabling mode-seeking fitting more effectively. In our experiments, we show that H-DPO outperformed DPO across various tasks, demonstrating superior results in pass@$k$ evaluations for mathematical tasks. Moreover, H-DPO is simple to implement, requiring only minor modifications to the loss calculation of DPO, which makes it highly practical and promising for wide-ranging applications in the training of LLMs.

## 1 Introduction

Large language models (LLMs) have exhibited remarkable performance across various tasks (OpenAI et al., 2023; Dubey et al., 2024). However, large datasets often include data created for various purposes, and the models trained on these datasets are not always suitable for users' specific needs. Additionally, some datasets include malicious text and code related to cyberattacks, posing risks of misuse by humans or the AI itself (Bender et al., 2021; Bai et al., 2022; Ji et al., 2023; Shevlane et al., 2023).

Reinforcement Learning from Human Feedback (RLHF) (Christiano et al., 2017; Bai et al., 2022) is an effective approach to make an LLM follow human instructions and suppressing undesired outputs. In RLHF, a reward model is trained based on data evaluated according to human preferences. The LLM then learns to maximize rewards, aligning its outputs with human preferences. To prevent significant deviation from the original model, regularization using reverse KL divergence is added to the reward maximization process, and RL algorithms such as PPO (Schulman et al., 2017) are employed.

However, RLHF has issues such as high computational costs, the reliance on a learned reward model, and the inherent instability and hyperparameter sensitivity of RL algorithms. To address these problems, Direct Policy Optimization (DPO) (Rafailov et al., 2023) has emerged and is now widely used. DPO proposes a loss function that directly optimizes the policy through a change of variables, eliminating the need for the reward model and allowing training with a simple binary cross-entropy loss. While more stable and lightweight than RLHF, DPO can optimize the same objective function as RLHF, which involves reward maximization and regularization with the reverse KL divergence. Other types of divergences have also been proposed to prevent deviation from the original model (Wang et al., 2024a), but reverse KL divergence, which enables mode-seeking estimation, is generally preferred for performance.

We point out that minimizing reverse KL divergence can cause the mode of the fitted distribution to fail to capture the mode of the target distribution. As shown in Figure 1, consider fitting a unimodal distribution to a multimodal distribution. We call the way of fitting a distribution *mode-seeking* when one of the modes of target distribution is captured by the fitted model as shown in the right side of Figure 1, and *mode-covering*

when all the modes are covered as shown in the left side of Figure 1. In the case of mode-seeking, the fitted distribution discards other modes of the target distribution, resulting in smaller variance than the target distribution. However, reverse KL minimization can fail at mode-seeking fitting due to its nature of preserving variance, as illustrated in the left side of Figure 1.

To enable variance reduction and encourage mode-seeking estimation, we generalize the loss function of DPO, named H-DPO, which allows for controlling the distribution's entropy $H(\pi)$ by modifying the regularization term. H-DPO can adjust the entropy of generations of the LLM during training using the hyperparameter $\alpha$ in Equation (9) introduced later. By setting $\alpha$ less than 1, it encourages the entropy to be reduced so that achieves mode-seeking fitting more successfully. The right side of Figure 1 demonstrates that our regularizer $D_\alpha$, a modification to the reverse KL, enables mode-seeking fitting even in cases where reverse KL fails, as shown on the left.

Using our proposed loss with $\alpha < 1$, the estimated policy distribution is expected to be sharper or more deterministic, which we consider a beneficial feature

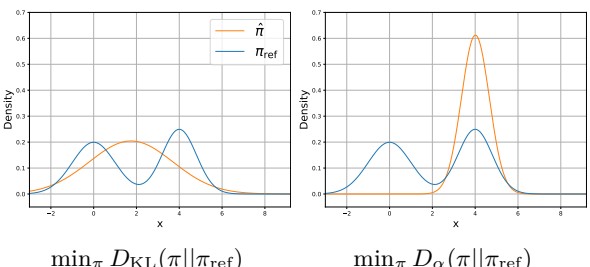

$$\min_\pi D_{\mathrm{KL}}(\pi||\pi_{\mathrm{ref}}) \qquad \min_\pi D_\alpha(\pi||\pi_{\mathrm{ref}})$$

Figure 1: For a Gaussian mixture model $\pi_{\mathrm{ref}}$, $\hat{\pi}$ that minimizes $D_{\mathrm{KL}}$ (left) and $\hat{\pi}$ that minimizes $D_\alpha = -\alpha H(\pi) + H(\pi, \pi_{\mathrm{ref}})$ with $\alpha = 0.6$ (right). Using $D_\alpha$ results in successful mode-seeking estimation.

rather than a problem. Traditional LLMs use a softmax function with a temperature parameter to represent distributions over raw outputs, where the temperature is set to 1 during training. When LLMs are evaluated, a lower value such as 0.6 often performs better (Xu et al., 2022; OpenAI et al., 2023; Zhu et al., 2024). This post-training sharpening lacks guarantees of optimality for the objective function. In contrast, our proposed method trains the language model using an objective function aimed at sharpening the distribution, ensuring that this sharper distribution aligns with the objective function.

Our main contribution is the alignment method H-DPO, which allows controlling entropy and encourages mode-seeking fitting more than DPO. The implementation of H-DPO is simple, requiring minimal modifications to DPO. Experiments included alignment based on Mistral-7B (Jiang et al., 2023) with the Zephyr framework (Tunstall et al.; 2023). Compared to DPO, our proposed method allows for more diverse generations without losing performance, and shows superior accuracy and coverage across various tasks.

## 2 Related Work

**Alignment** Language models trained through next-token prediction have rapidly advanced and show strong performance on many tasks in zero-shot or few-shot settings (Radford et al., 2019; Brown et al., 2020; Chowdhery et al., 2023). Fine-tuning using human preferences and instructions, known as alignment, has proven effective in improving instruction following and reducing harmful outputs (Christiano et al., 2017; Bai et al., 2022; Touvron et al., 2023; Ouyang et al., 2022). A prominent method for alignment is RLHF; however, it encounters issues such as high computational costs, significant memory requirements, and the instability of reinforcement learning (Schulman et al., 2017; Engstrom et al., 2020; Ahmadian et al., 2024). To address these issues, DPO (Rafailov et al., 2023) has been proposed. DPO eliminates the need to model the reward function and employ reinforcement learning algorithms, evolving in various directions (Liu et al., 2024; Gheshlaghi Azar et al., 2024; Song et al., 2024). Wang et al. (2024a); Zeng et al. (2024) enable adjusting the diversity of generated responses by changing the regularization of reverse KL. Wang et al. (2024a) extends this to $f$-divergence other than reverse KL divergence, arguing that adjusting $\alpha$ in $\alpha$-divergence allows for a trade-off between diversity and performance. As $\alpha$-divergence interpolates between reverse KL and forward KL, using larger $\alpha$ makes the mode-seeking property diminish, which may increase diversity but deteriorate performance. Our study proposes a different method to balance diversity and performance while maintaining or strengthening the mode-seeking property of reverse KL.

**Diversity in Language Models** The importance of diversity in the responses generated by language models has been emphasized in numerous studies. Achieving more diverse text generation with high quality is crucial, despite the existing quality-diversity trade-off (Nenkova et al., 2007; Clarke et al., 2008; Hashimoto et al., 2019; Zhang et al., 2021). Diversity can be adjusted through various methods, such as sampling-based techniques like changing the temperature (Fan et al., 2018; Holtzman et al., 2020; Wang et al., 2024b), manipulating prompts (Arora et al., 2023; Li et al., 2023), or during DPO as mentioned in the previous section (Wang et al., 2024a; Zeng et al., 2024). Studies such as Wang et al. (2024a); Zeng et al. (2024) have examined changes in diversity due to objective functions in post-training, but have not considered the impact of temperature adjustments, which are commonly manipulated when using language models. Our study investigates the effects of both objective function modifications in post-training and temperature adjustments on diversity.

In recent LLMs, there has been a growing emphasis not only on the accuracy of a single response but also on the coverage — the fraction of problems solved by any generated sample (Kulal et al., 2019; Chen et al., 2021; Roziere et al., 2023; Brown et al., 2024). In such evaluations, diversity in the generated outputs contributes to improved coverage (Wang et al., 2024b). The importance of coverage is partly due to the presence of verifiers that can assess the correctness of generated answers, particularly in mathematical and coding tasks. These verifiers allow for selecting correct outputs from multiple candidates as the answer. Some studies (Kulal et al., 2019; Chen et al., 2021; Roziere et al., 2023) have demonstrated significant improvements in correctness through repeated sampling in coding tasks, while Brown et al. (2024) showed that even relatively lightweight models could outperform frontier models in coverage by increasing the number of generated samples in mathematical tasks. In tasks such as chat, where precise verification is challenging, the performance can still be enhanced through methods such as majority voting (Wang et al., 2023) or by using reward models and trained verifiers (Cobbe et al., 2021; Lightman et al., 2024; Hosseini et al., 2024; Wang et al., 2024c; Kang et al., 2024) to use repeated samples effectively.

Wang et al. (2024b) explored the relationship between diversity and coverage, demonstrating that greater diversity in generated outputs leads to a more significant improvement in coverage for larger values of $k$ in pass@$k$, which denotes the probability that the correct answer is included in the $k$ generated outputs. Our study shows that using the proposed objective can increase diversity while maintaining a certain level of accuracy, achieving favorable performance in pass@$k$ evaluations.

**Mode-Seeking and Mode-Covering** When minimizing a certain divergence to bring two probability distributions closer, attention is often given to whether the fitting or divergence is mode-seeking or mode-covering (mass-covering) (Huszár, 2015; Shannon et al., 2020; Ke et al., 2021; Li & Farnia, 2023; Wang et al., 2024a). When fitting a distribution to a multimodal distribution, if the fitted distribution captures one of the modes, this fitting is called mode-seeking. If it covers all the modes, it is termed mode-covering. Accordingly, divergences facilitating such fittings when minimized are similarly referred to as mode-seeking and mode-covering divergences, respectively. The reverse KL divergence, which is used in RLHF and DPO training, is considered mode-seeking compared to forward KL and other $f$-divergence Shannon et al. (2020); Li & Farnia (2023). Policy learning using mode-seeking divergence often performs better than mode-covering divergence (Ke et al., 2021; Wang et al., 2024a). In this study, we propose a new regularizer to replace the minimization of reverse KL divergence in the objective function of DPO, aiming to achieve better performance through enhanced mode-seeking property.

## 3 Preliminaries

### 3.1 Reinforcement Learning from Human Feedbacks (RLHF)

In the context of LLM training, RLHF is a process of aligning an LLM to human preferences after pre-training, typically consisting of three steps: supervised fine-tuning (SFT), reward modeling, and RL fine-tuning.

**Supervised Fine-Tuning (SFT)** SFT is the process of adapting an already pre-trained LLM to specific tasks by optimizing the model parameters using a task-specific dataset. Using high-quality data related to the task, the model is optimized through supervised learning to obtain $\pi_{\text{SFT}}$.

**Reward Modeling**   Next, a reward model is trained to reflect human preferences in RL. Let $r_\phi(x, y)$ be a reward model parameterized by $\phi$, where $x$ is a prompt and $y$ is a completion. It is typically assumed that human preference for a pair of completions follows the Bradley-Terry (BT) model (Bradley & Terry, 1952), where the probability of preferring $y_1$ to $y_2$ is represented using a difference of rewards:

$$p(y_1 \succ y_2 \mid x) = \sigma(r(x, y_1) - r(x, y_2)), \tag{1}$$

where $\sigma(x) = \frac{1}{1+\exp(-x)}$ is a sigmoid function. The larger the value of $r(x, y)$ is, the more preferable a completion $y$ is to a prompt $x$.

Using a labeled dataset $\mathcal{D} = \{x^i, y_w^i, y_l^i\}_{i=0}^N$ with user preferences, where $y_w^i$ is preferred to $y_l^i$ for prompt $x^i$, the loss function for training the reward model is formulated by minimizing the negative log-likelihood:

$$L(r_\phi) = -\mathbb{E}_{(x, y_w, y_l \sim \mathcal{D})}[\log \sigma(r_\phi(x, y_w) - r_\phi(x, y_l))]. \tag{2}$$

**RL Fine-Tuning**   Finally, the language model is fine-tuned, using the trained reward model, to maximize the following objective function:

$$J(\pi_\theta) = \mathbb{E}_{x \sim \mathcal{D}, y \sim \pi_\theta}[r_\phi(x, y)] - \beta D_{\mathrm{KL}}(\pi_\theta(y \mid x) \| \pi_{\mathrm{ref}}(y \mid x)), \tag{3}$$

where $\beta$ is a hyperparameter that controls the deviation from $\pi_{\mathrm{ref}}$. $\pi_\theta$ is trained to maximize the reward while being regularized by the reverse KL divergence to not deviate too much from $\pi_{\mathrm{ref}}$. Typically, $\pi_{\mathrm{ref}}$ is fixed to $\pi_{\mathrm{SFT}}$ while $\pi_\theta$ is initialized with $\pi_{\mathrm{SFT}}$.

### 3.2   Directed Preference Optimization (DPO)

In RLHF, the need to train the reward model and apply an online RL algorithm such as PPO imposes significant computational and memory costs. DPO suggests a method for directly learning to reflect human preferences in a supervised manner without using the reward model by mapping language model policies and reward functions. The objective function is equivalent to that of RLHF, and the optimal policy that maximizes Equation (3) when the reward model is optimal is derived as follows:

$$\pi^*(y \mid x) = \frac{1}{Z(x)} \pi_{\mathrm{ref}}(y \mid x) \exp\left(\frac{r^*(x, y)}{\beta}\right), \tag{4}$$

where $Z(x)$ is the partition function. From this equation, the optimal reward can be expressed using the optimal policy:

$$r^*(x, y) = \beta \log \frac{\pi^*(y \mid x)}{\pi_{\mathrm{ref}}(y \mid x)} + \beta \log Z(x). \tag{5}$$

Using this optimal reward function to calculate the probability distribution of the BT model, the computationally challenging partition function $Z(x)$ cancels out as follows:

$$p^*(y_1 \succ y_2 \mid x) = \sigma\left(\beta \log \frac{\pi^*(y_1 \mid x)}{\pi_{\mathrm{ref}}(y_1 \mid x)} - \beta \log \frac{\pi^*(y_2 \mid x)}{\pi_{\mathrm{ref}}(y_2 \mid x)}\right). \tag{6}$$

The loss function for $\pi_\theta$ is derived as the maximum likelihood estimation of the BT model from a human preference dataset $\mathcal{D}$:

$$L_{\mathrm{DPO}} = -\mathbb{E}_{x, y_w, y_l \sim \mathcal{D}}\left[\log \sigma\left(\beta \log \frac{\pi_\theta(y_w \mid x)}{\pi_\theta(y_l \mid x)} - \beta \log \frac{\pi_{\mathrm{ref}}(y_w \mid x)}{\pi_{\mathrm{ref}}(y_l \mid x)}\right)\right]. \tag{7}$$

Thus, DPO can align language models with human preferences without learning a reward model.

## 4 Entropy Controllable Directed Preference Optimization

In DPO, reverse KL divergence is used as a regularizer that controls the deviation from $\pi_{\mathrm{ref}}$. The reverse KL divergence is defined as $D_{\mathrm{KL}}(\pi_\theta||\pi_{\mathrm{ref}}) = \int \pi_\theta(y \mid x) \log \frac{\pi_\theta(y|x)}{\pi_{\mathrm{ref}}(y|x)} dy$. Here, the integrand is zero for regions where $\pi_\theta(y \mid x) = 0$, meaning that only the regions supported by $\pi_\theta(y \mid x)$ affect the divergence. Consequently, fitting by minimizing the reverse KL divergence is known to be mode-seeking and generally performs better than other divergences such as forward KL (Ke et al., 2021; Wang et al., 2024a).

However, in this study, we discuss cases where even using reverse KL divergence can fail to achieve mode-seeking fitting with respect to the target distribution. We verify such cases through preliminary experiments and show that controlling the entropy of the distribution enables more effective mode-seeking fitting. To control the entropy of the output probability by language models in DPO, we propose H-DPO, which incorporates such entropy-controllable optimization into DPO.

### 4.1 Mode-seeking Property

As a preliminary experiment on the mode-seeking property of reverse KL divergence, we fit a Gaussian distribution to a mixture of two Gaussian components. Specifically, given a Gaussian mixture model $\pi_{\mathrm{ref}}$, we compute the location and scale parameters of a Gaussian distribution $\pi$ that minimize the reverse KL divergence $D_{\mathrm{KL}}(\pi||\pi_{\mathrm{ref}})$. If the fitting is mode-seeking, the estimated Gaussian distribution should capture one of the components of the mixture model. However, as shown in Figure 1, despite the reverse KL minimization, which is supposed to have the mode-seeking property, the fitting may look mode-covering, not mode-seeking. In this case, if $\pi$ is a language model, it is likely to generate from valleys where $\pi_{\mathrm{ref}}$ has a low probability, possibly leading to degraded performance of $\pi$.

The cause of such mode-covering fitting could be the inherent property of reverse KL divergence minimization, which aims to preserve some variance. If $\pi$ captures only one component, its variance should be smaller compared to $\pi_{\mathrm{ref}}$ as a whole because it must ignore the other component. As shown on the left side of Figure 1, however, reverse KL minimization does not take this into account, resulting in mode-covering estimation.

We consider an objective that can reduce variance or entropy as a remedy. To adjust the entropy of $\pi$, we note that the reverse KL divergence can be decomposed into entropy and cross-entropy components as follows:

$$
\begin{aligned}
D_{\mathrm{KL}}(\pi||\pi_{\mathrm{ref}}) &= \int (\pi(x) \log \pi(x) - \pi(x) \log \pi_{\mathrm{ref}}(x)) dx \\
&= -H(\pi) + H(\pi, \pi_{\mathrm{ref}}).
\end{aligned}
\tag{8}
$$

By attaching a coefficient $\alpha$ to the entropy $H(\pi)$, we can derive another objective that can control entropy: $D_\alpha = -\alpha H(\pi) + H(\pi, \pi_{\mathrm{ref}})^\dagger$. By making $\alpha$ less than 1, we can reduce the entropy while fitting between distributions. The right side of Figure 1 shows the distribution $\pi$ that minimizes $D_\alpha$ as $\alpha$ decreases from 1. By reducing $\alpha$ from 1 to a smaller value, it can achieve the mode-seeking fitting. Details of the preliminary experiments related to Figure 1 are provided in Appendix A.1.

The effectiveness of the mode-seeking property has been verified in Wang et al. (2024a), and strengthening the mode-seeking property by reducing $\alpha$ is an attractive feature. However, even in cases where $\pi$ and $\pi_{\mathrm{ref}}$ have the same number of modes (e.g., when both are unimodal distributions), allowing $\pi$ to fit $\pi_{\mathrm{ref}}$ with $D_\alpha$ can result in $\pi$ becoming a sharper distribution than $\pi_{\mathrm{ref}}$. Although this might seem problematic, it could be beneficial in language model training. For better performance at inference time, the sampling temperature is often set below 1 (Xu et al., 2022; OpenAI et al., 2023; Zhu et al., 2024). This means the distribution learned at a temperature of 1 is sharpened by reducing the temperature. However, there is no guarantee that the sharpened distribution is optimal for the DPO objective function. The distribution learned by maximizing our objective function with a small $\alpha$ also becomes sharp, but unlike adjusting sampling temperature at inference time, it becomes sharp in a manner consistent with the objective function. The following section introduces how to incorporate such entropy adjustment using $\alpha$ into DPO.

---

$^\dagger$Note that, for $\alpha \neq 1$, $D_\alpha(p||q)$ is not a divergence because it is not zero even when $p = q$.

## 4.2 H-DPO

As discussed in the previous section, by decomposing the reverse KL divergence into its entropy and cross-entropy components, we can adjust the entropy with $\alpha$. The objective function for DPO with entropy adjustment is shown below:

$$
\begin{aligned}
J_{\text{H-DPO}} &= \mathbb{E}_{x\sim\mathcal{D},y\sim\pi}\left[r(x,y)\right] - \beta D_\alpha(\pi||\pi_{\text{ref}}) \\
&= \mathbb{E}_{x\sim\mathcal{D},y\sim\pi}\left[r(x,y)\right] + \alpha\beta H(\pi) - \beta H(\pi,\pi_{\text{ref}}).
\end{aligned}
\tag{9}
$$

Here, when $\alpha$ equals 1, it becomes the same objective function as that of standard DPO. By setting $\alpha$ to be smaller than 1, the learning process aims to reduce the entropy. Similar to Wang et al. (2024a), we consider a constrained optimization. By applying Lagrange multipliers under the constraints that $\pi$ is a probability distribution, i.e., $\sum_y \pi(y \mid x) = 1$ and $\forall y, \pi(y \mid x) \geq 0$, we obtain the following:

$$
\mathcal{L}(\pi,\lambda,C) = \mathbb{E}_{x\sim\mathcal{D},y\sim\pi}[r(x,y) - \alpha\beta\log\pi(y \mid x) + \beta\log\pi_{\text{ref}}(y \mid x)] - \lambda\left(\sum_y \pi(y \mid x) - 1\right) - \sum_y C(y)\pi(y \mid x),
\tag{10}
$$

where $\lambda$ and $C$ are the dual variables. Solving this problem, the optimal policy $\pi^*$ can be derived as

$$
\pi^*(y \mid x) = \frac{1}{Z(x)}\pi_{\text{ref}}(y \mid x)^{1/\alpha}\exp\left(\frac{r^*(x,y)}{\alpha\beta}\right).
\tag{11}
$$

From this equation, the reward function can be expressed using the policy as follows:

$$
r^*(x,y) = \alpha\beta\log\pi^*(y \mid x) - \beta\log\pi_{\text{ref}}(y \mid x) + \alpha\beta\log Z(x).
\tag{12}
$$

When applying this reward function to the BT model and performing the maximum likelihood estimation, the loss function using $\alpha$ is

$$
L_{\text{H-DPO}} = -\mathbb{E}_{x,y_w,y_l\sim\mathcal{D}}\left[\log\sigma\left(\alpha\beta\log\frac{\pi_\theta(y_w \mid x)}{\pi_\theta(y_l \mid x)} - \beta\log\frac{\pi_{\text{ref}}(y_w \mid x)}{\pi_{\text{ref}}(y_l \mid x)}\right)\right].
\tag{13}
$$

Comparing this equation to the DPO loss function in Equation (7), we can see that entropy adjustment using $\alpha$ can be implemented by simply replacing the coefficient for $\pi_\theta$ from $\beta$ to $\alpha\beta$.

## 5 Experiments

In this section, we evaluate the performance of H-DPO in comparison to standard DPO using widely recognized metrics.

### 5.1 Experimental Setup

We conducted DPO training based on Zephyr-7B-Beta (Tunstall et al., 2023; Tunstall et al.). We started from zephyr-7b-sft-full, which is based on Mistral 7B (Jiang et al., 2023) and fine-tuned with UltraChat (Ding et al., 2023). We performed DPO training on it with UltraFeedback (Cui et al., 2023). We evaluated the performance when H-DPO was used instead of standard DPO. The hyperparameters during training were the same as those of Zephyr-7B-beta, except for the variable $\alpha$. The $\alpha$ was varied in the range from 0.8 to 1.2. Another model, Llama-3.2-1B (Dubey et al., 2024), was also used for the experiments, and the results are detailed in Appendix A.3.

The evaluation tasks included diverse grade school math word problems (GSM8K (Cobbe et al., 2021)), coding task (HumanEval (Chen et al., 2021)), multiple-choice question task (MMLU-Pro (Wang et al., 2024d)) and instruction-following task (IFEval (Zhou et al., 2023)). The training was conducted with three different seeds. Further experimental details are provided in Appendix A.4 and A.5.

## 5.2 Performance and Diversity

Table 1 shows the scores for each task when $\alpha$ was decreased. By reducing $\alpha$ by 0.05 to 0.1, performance improved on all tasks compared to the conventional DPO ($\alpha = 1$).

Table 2 presents diversity metrics when $\alpha$ was varied in H-DPO. When the temperature was set to 1, smaller $\alpha$ values resulted in lower diversity, while larger $\alpha$ values increased diversity. This indicates that diversity can be controlled through $\alpha$. However, it should be noted that diversity changes with temperature, and the optimal temperature varies depending on the value of $\alpha$. Hence, even with a smaller $\alpha$, diversity could be increased if a higher temperature is used.

For MMLU-Pro, the scores and entropy with varying temperatures are shown in Figure 2. The left figure illustrates the relationship between temperature and score, highlighting that smaller $\alpha$ values exhibit less performance degradation and greater robustness to temperature selection. This is because entropy remains low even when a higher temperature is used. The right figure shows the relationship between entropy and score, where the entropy of the samples obtained at each temperature replaces the temperature shown in the left figure. At the same score point, the entropy is larger when $\alpha$ is smaller. In other words, with a smaller $\alpha$, it is possible to achieve more diverse generations even with the same performance.

Table 1: Average scores of DPO and H-DPO with different $\alpha$ values on various tasks.

| | GSM8K↑ | HumanEval↑ | MMLU-Pro↑ | IFEval↑ |
|---|---|---|---|---|
| DPO ($\alpha = 1$) | 26.40 $_{\pm1.76}$ | 28.77 $_{\pm0.45}$ | 31.83 $_{\pm0.17}$ | 59.63 $_{\pm0.72}$ |
| H-DPO ($\alpha = 0.95$) | 27.77 $_{\pm1.39}$ | **30.70** $_{\pm0.39}$ | **32.37** $_{\pm0.03}$ | 60.17 $_{\pm0.34}$ |
| H-DPO ($\alpha = 0.9$) | **28.83** $_{\pm2.32}$ | 29.63 $_{\pm0.45}$ | 32.30 $_{\pm0.17}$ | **60.93** $_{\pm0.50}$ |
| H-DPO ($\alpha = 0.8$) | 28.66 $_{\pm1.23}$ | 27.77 $_{\pm0.67}$ | 31.93 $_{\pm0.19}$ | 59.90 $_{\pm0.59}$ |

Table 2: Comparison of DPO and H-DPO with various $\alpha$ values across different diversity metrics when temperature is 1.

| | Entropy↑ | Self-Bleu↓ | Distinct-1↑ | Distinct-2↑ |
|---|---|---|---|---|
| H-DPO ($\alpha = 1.2$) | 1.718 | 0.252 | 0.313 | 0.690 |
| H-DPO ($\alpha = 1.1$) | 1.483 | 0.293 | 0.296 | 0.652 |
| DPO ($\alpha = 1$) | 1.323 | 0.326 | 0.289 | 0.633 |
| H-DPO ($\alpha = 0.95$) | 1.223 | 0.339 | 0.277 | 0.611 |
| H-DPO ($\alpha = 0.9$) | 1.113 | 0.364 | 0.272 | 0.590 |
| H-DPO ($\alpha = 0.8$) | 0.977 | 0.391 | 0.268 | 0.574 |

## 5.3 Coverage Evaluation

As mentioned in the previous section, a smaller $\alpha$ enabled more diverse outputs at the same performance level. Wang et al. (2024b) demonstrated that high diversity positively impacts coverage performance, where coverage is evaluated using the pass@$k$ metric. Coverage refers to the fraction of problems that can be solved using any generated sample, and pass@$k$ is the coverage achieved by using $k$ samples (Kulal et al., 2019; Chen et al., 2021). Chen et al. (2021) proposed an unbiased and stable calculation method for pass@$k$ metric, which is employed in our study. Coverage is particularly significant in tasks where correctness evaluation is relatively straightforward, such as mathematical and coding tasks; hence, evaluations were conducted on the GSM8K (math task) and HumanEval (coding task).

Figure 3 presents the pass@$k$ evaluation results for various $k$ values in GSM8K. Overall, reducing $\alpha$ leads to better performance than standard DPO ($\alpha = 1$). In standard DPO, for most values of $k$, the best coverage when varying the temperature is achieved at a temperature of 0.5, which is smaller than the value of 1 used during training. However, for smaller $\alpha$ values (e.g., $\alpha = 0.8$), the best coverage is achieved with the same training temperature of 1 when $k$ is large. This implies that decreasing $\alpha$ ($\alpha = 0.8$) and using a temperature close to that used during training provides better results than simply lowering the temperature in standard

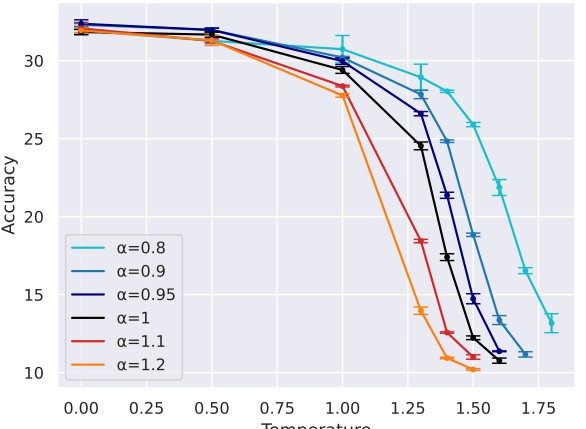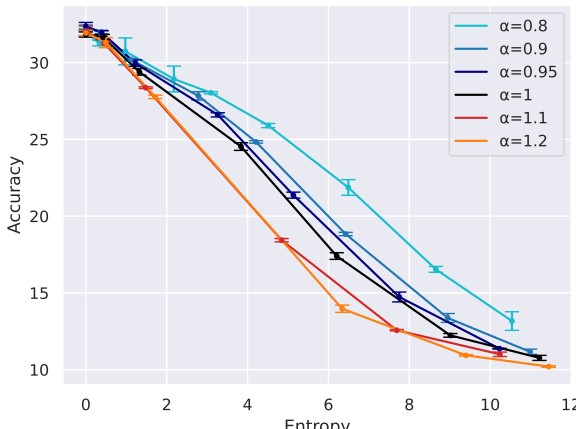

Figure 2: **Left**: Accuracy on **MMLU-Pro** at various temperatures. **Right**: Accuracy on **MMLU-Pro** at various entropy levels. The horizontal axis of the left figure is replaced with the entropy obtained from sampling at each corresponding temperature.

DPO. This suggests that H-DPO, which allows using a model closer to the one used during training even at test time, is superior to standard DPO in this setting.

Figure 4 presents the evaluation results of pass@$k$ for various values of $k$ on the HumanEval benchmark. On HumanEval, there is a negligible difference between models with a small $\alpha$ and standard DPO when $k$ is large. However, interestingly, when $k$ exceeds 100, the results improve for larger $\alpha$ values ($\alpha = 1.1$).

## 5.4 Discussion

In the evaluation of the HumanEval coding task and GSM8K mathematical task, we observed that the optimal values of $\alpha$ differed between these two task categories. This discrepancy can be attributed to differences in task characteristics, which necessitate distinct sampling temperatures for effective generation. In mathematical tasks, where there is a single correct answer and precise reasoning is required, more deterministic sampling with a lower temperature is preferable. In these cases, values of $\alpha$ less than 1 are suited, facilitating more precise generations. Conversely, in coding tasks, multiple valid answers typically exist, and generating diverse outputs increases the likelihood of producing correct responses. As a result, a sampling temperature of 1 is more suitable for pass@$k$ evaluations in such scenarios. Note that when the temperature exceeds the training value of 1, a significant decline in performance is observed. In such cases, values of $\alpha$ greater than 1 further enhance diversity, as shown in Table 2, improving the probability of generating correct responses in pass@$k$ evaluations.

In summary, for tasks requiring accuracy and utilizing a temperature lower than 1, an $\alpha$ value slightly less than 1, such as 0.9 or 0.95, is appropriate. Conversely, for tasks emphasizing diversity and employing a temperature of 1, using an $\alpha$ value greater than 1, such as 1.1, yields better results.

As suggested by Figure 3 and 4, a practical approach to tuning the $\alpha$ parameter is to first train the model using the standard DPO setting ($\alpha = 1$) and then evaluate the performance changes by varying the temperature. For HumanEval with smaller $k$ values and GSM8K, performance improves when the temperature is slightly reduced from 1, indicating that more accurate outputs are preferable, and this improvement aligns with lowering $\alpha$. Conversely, for HumanEval with larger $k$ values, performance degrades as the temperature decreases from 1, suggesting that diversity is critical in such cases, which explains the relatively better performance with $\alpha > 1$. In this way, $\alpha$ tuning can be guided by observing whether performance improves or declines when the temperature deviates from 1.

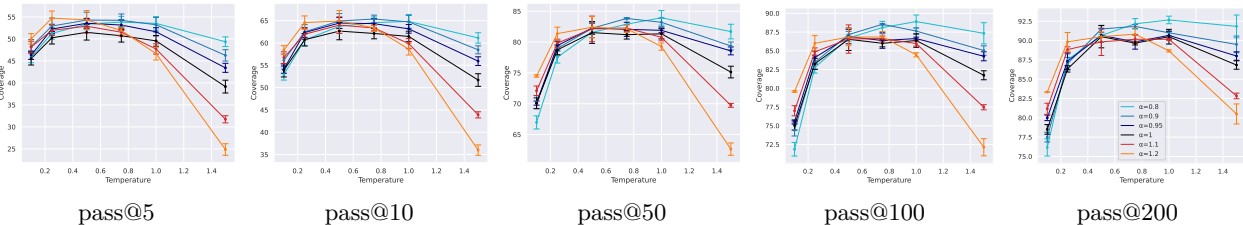

pass@5 pass@10 pass@50 pass@100 pass@200

Figure 3: Coverage (pass@$k$) of H-DPO and DPO with various temperatures on **GSM8K**.

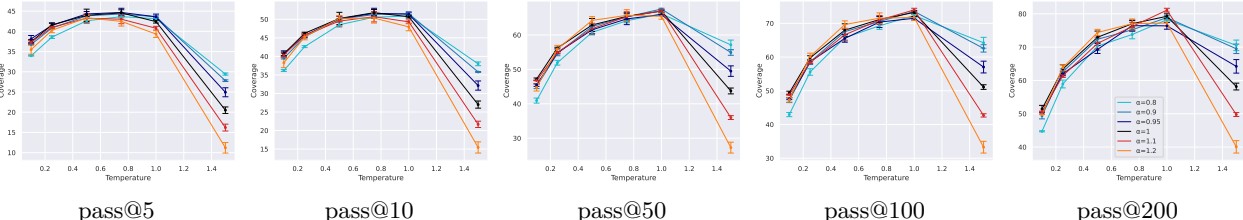

pass@5 pass@10 pass@50 pass@100 pass@200

Figure 4: Coverage (pass@$k$) of H-DPO and DPO with various temperatures on **HumanEval**.

## 6 Conclusion

In this study, we proposed H-DPO, a generalization of DPO, and thoroughly examined its effectiveness. H-DPO allows for the adjustment of entropy during training through the hyperparameter $\alpha$, enabling the control of distribution sharpness and achieving more effective mode-seeking fitting compared to standard DPO. This new method allows trained models to generate more accurate and diverse outputs, better aligning with their intended purposes. In the experiments, we aligned Mistral-7B-based models using the proposed method and compared them with standard DPO. H-DPO demonstrated superior performance compared to DPO across various tasks. In mathematical tasks, it showed excellent performance in pass@$k$ evaluations. These results confirmed that the diversity and quality of the generated outputs improved, establishing H-DPO as a powerful method for improving the training process of LLMs. Moreover, H-DPO is extremely simple to implement, requiring only minor modifications to existing DPO, which adds to its practicality and potential for widespread application. The need to adjust $\alpha$ is a limitation of this method, and automating the search of appropriate $\alpha$ values for each task can be a focus of future research.

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

# A Experimental Details

## A.1 Preliminary Experiment with Gaussian Distribution

This section details the experiments shown in Figure 1. In this preliminary experiment, we use the proposed regularization $D_\alpha(\pi||\pi_{\text{ref}})$, where $D_\alpha$ is the same as the Kullback-Leibler divergence $D_{\text{KL}}$ when $\alpha = 1$, to estimate the Gaussian distribution $\pi$ that is closest to the Gaussian Mixture Model (GMM) $\pi_{\text{ref}}$. The experiments were conducted with GMMs $\pi_{\text{ref}}$ consisting of 2, 3, and 4 Gaussian components, and the results are shown in Figure 5, 7 and 8, respectively. For any $\pi_{\text{ref}}$, the weights of the components are equal, and the standard deviations are 1 and 0.8 for the case of two components, 1, 0.8, and 0.5 for the case of three components, and 1, 0.8, 0.5, and 0.3 for the case of four components. In those figures, the results of varying the interval between the means of the Gaussian components are displayed in separate rows.

In the upper row of Figure 5, we observe that when $\alpha = 1$, i.e. using the KL divergence, the fitting becomes mode-covering. When $\alpha$ is reduced to 0.6, it successfully achieves mode-seeking fitting. In the middle row, where the interval between the means of the components is larger, making mode-seeking fitting more feasible, mode-seeking fitting is observed at $\alpha = 0.8$. In the bottom row, where the interval is even larger, mode-seeking fitting occurs even when minimizing the KL divergence, although the fitting targets the Gaussian with the larger variance on the left. As $\alpha$ decreases, the fitting shifts to the Gaussian on the right, which has smaller variance and higher probability.

In Figure 6, the values of $D_\alpha(\pi||\pi_{\text{ref}})$ are represented using color as the location and scale parameters of the Gaussian distribution $\hat{\pi}$ are varied. As $\alpha$ decreases, the $D_\alpha(\pi||\pi_{\text{ref}})$ values for mode-seeking $\hat{\pi}$ become smaller compared to those for mode-covering $\hat{\pi}$.

Similar results are observed in Figure 7 and 8 for cases with 3 and 4 Gaussian components. When minimizing the KL divergence, the fitting tends to be mode-covering or targets the component with larger variance. However, reducing $\alpha$ results in the fitting successfully targeting the region with the highest probability in all cases. As $\alpha$ decreases further, the variance of $\hat{\pi}$ also becomes smaller.

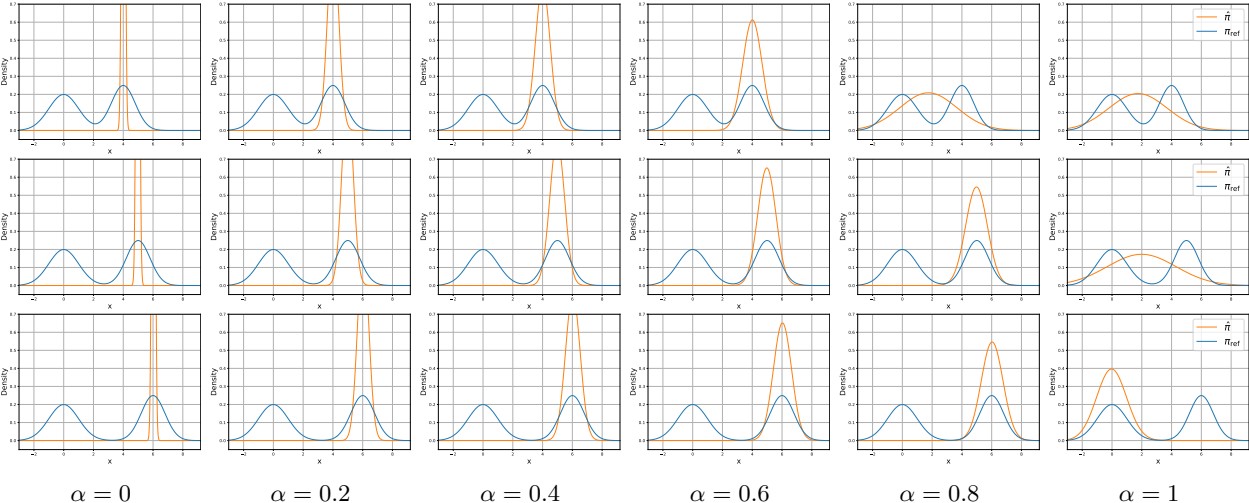

Figure 5: $\pi_{\mathrm{ref}}$ is a GMM composed of two normal distributions, and $\hat{\pi}$ represents the normal distribution that minimizes $D_{\alpha}(\pi||\pi_{\mathrm{ref}})$. The upper, middle, and bottom rows correspond to cases where the mean intervals between components are 4, 5, and 6, respectively. The standard deviations of each component are 1 and 0.8 from left to right.

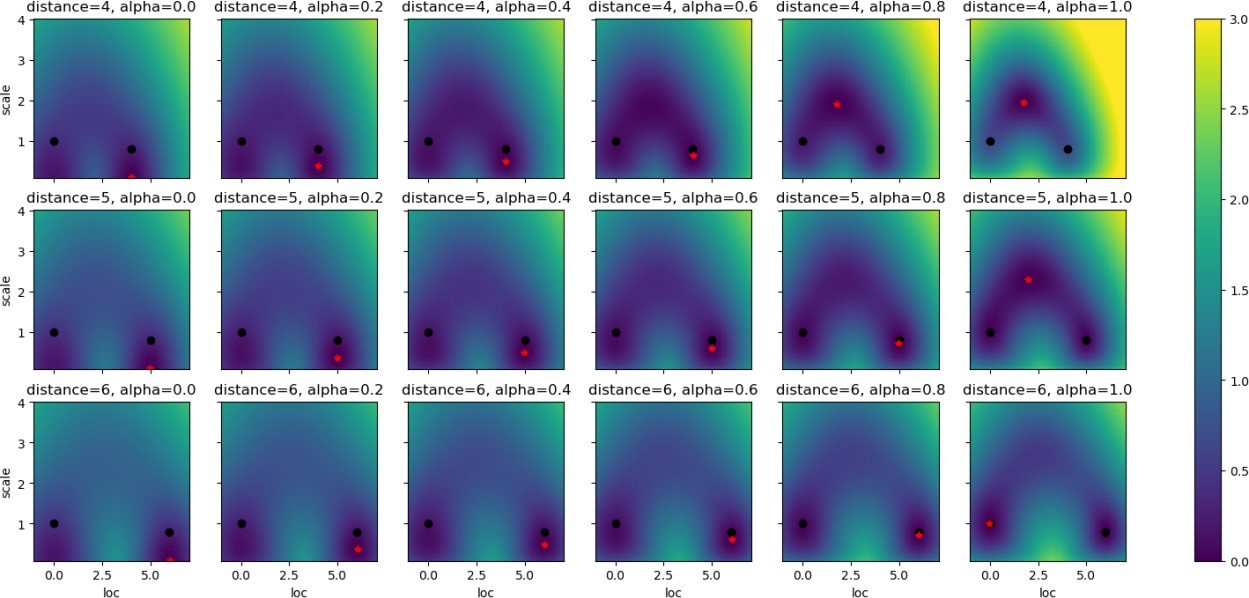

Figure 6: Values of $D_{\alpha}(\pi||\pi_{\mathrm{ref}})$ for the normal distribution $\pi$ with various location and scale parameters in the experiment shown in Figure 5. For visibility, $\min(3, \ln D_{\alpha}(\pi||\pi_{\mathrm{ref}}) - \ln D_{\alpha}(\hat{\pi}||\pi_{\mathrm{ref}}))$ is plotted. The red star indicates the parameters of $\hat{\pi}$ that minimize $D_{\alpha}(\pi||\pi_{\mathrm{ref}})$, and these values are used to plot Figure 5.

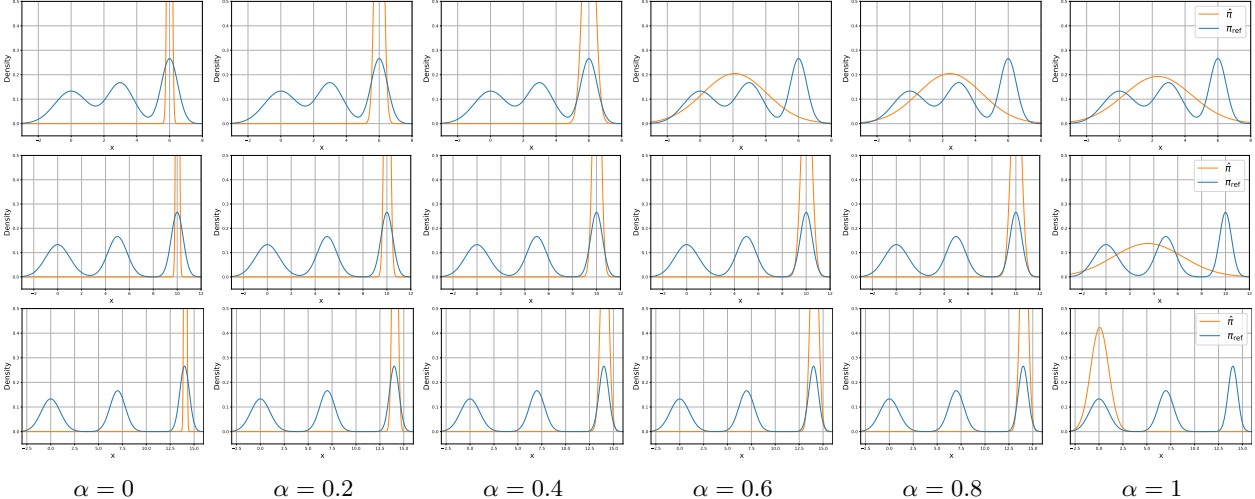

Figure 7: $\pi_{\text{ref}}$ is a GMM composed of two normal distributions, and $\hat{\pi}$ represents the normal distribution that minimizes $D_\alpha(\pi||\pi_{\text{ref}})$. The upper, middle, and bottom rows correspond to cases where the mean intervals between components are 3, 5, and 7, respectively. The standard deviations of each component are 1, 0.8 and 0.5 from left to right.

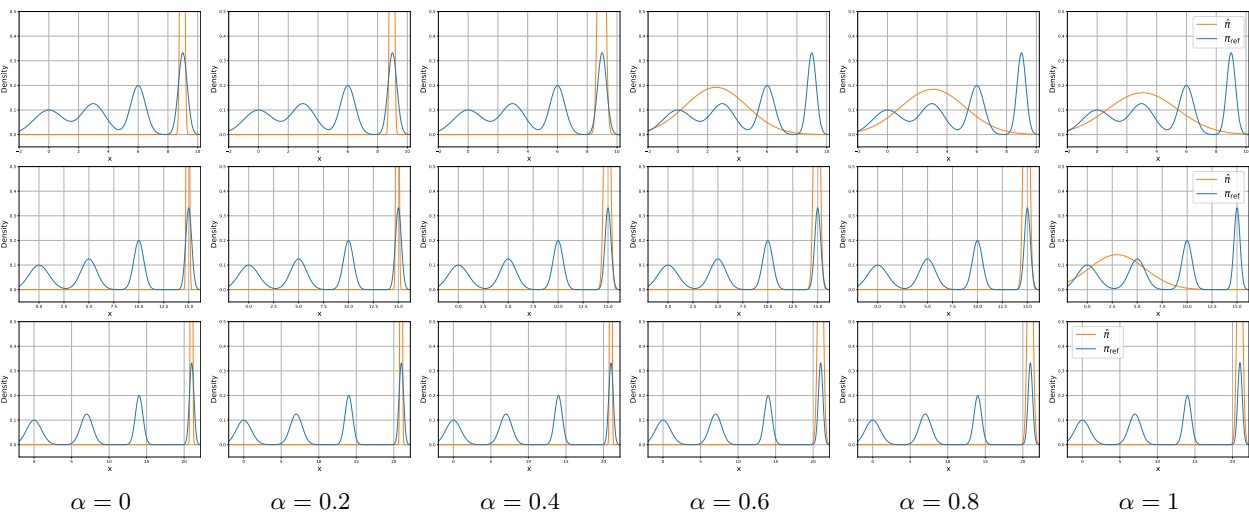

Figure 8: $\pi_{\text{ref}}$ is a GMM composed of two normal distributions, and $\hat{\pi}$ represents the normal distribution that minimizes $D_\alpha(\pi||\pi_{\text{ref}})$. The upper, middle, and bottom rows correspond to cases where the mean intervals between components are 3, 5, and 7, respectively. The standard deviations of each component are 1, 0.8, 0.5 and 0.3 from left to right.

## A.2  Comparison with $\beta$ tuning

From Table 1, we observed that performance improves by decreasing the value of $\alpha$. This raised the possibility that similar improvements might be achievable by tuning the $\beta$ parameter in standard DPO. Therefore, we compared the performance of H-DPO, which showed promising results with parameters $(\alpha = 0.9, \beta = 0.01)$, against a DPO where $\beta$ was similarly reduced $(\alpha = 1, \beta = 0.009)$. The results are presented in Table 4, showing that tuning $\beta$ in DPO does not achieve the same level of improvement as H-DPO. The accuracy decreased in many tasks. The evaluation of coverage on the GSM8K dataset is shown in Figure 9, which indicates that tuning $\beta$ in DPO does not improve coverage either. These results suggest that the performance enhancement obtained by tuning $\alpha$ to adjust the entropy cannot be replicated through $\beta$ adjustment in DPO, thus demonstrating the effectiveness of H-DPO.

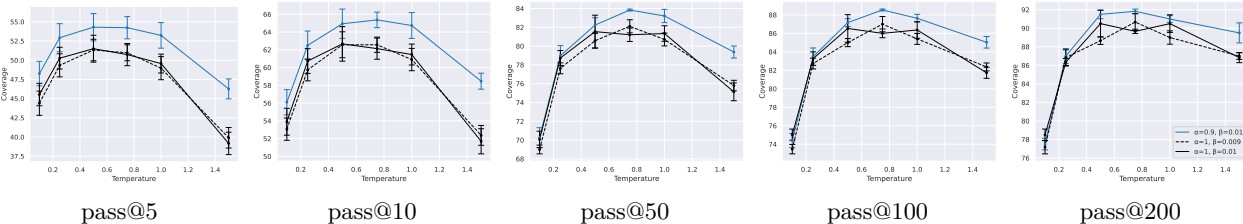

| pass@5 | pass@10 | pass@50 | pass@100 | pass@200 |

Figure 9: Coverage (pass@$k$) of H-DPO and DPO with various temperatures on **GSM8K**.

## A.3  Experiments with Llama-3.2-1B

To further demonstrate the applicability of H-DPO across different models, we conducted experiments using Llama-3.2-1B (Dubey et al., 2024). To differentiate these experiments from those performed with Zephyr, we utilized a different dataset, the Anthropic HH dataset (Bai et al., 2022). The experimental setup was consistent with Rafailov et al. (2023), where Llama-3.2-1B underwent SFT using only the preference completions from the dataset, followed by fine-tuning with H-DPO. The value of $\beta$ was set to 0.01, while all other hyperparameters matched those used in Rafailov et al. (2023).

The results of the experiments conducted on four tasks are presented in Table 3. Given the difficulty of the tasks and the inherently low performance of the base model, consistent improvements were not observed in HumanEval. However, we did observe performance improvements in other tasks.

Table 3: Average scores of DPO and H-DPO with different $\alpha$ values on various tasks when using the Llama-3.2-1B model.

| | GSM8K↑ | HumanEval↑ | MMLU-Pro↑ | IFEval↑ |
|---|---|---|---|---|
| DPO ($\alpha = 1$) | 4.97 $_{\pm 0.31}$ | **2.73** $_{\pm 1.43}$ | 14.20 $_{\pm 0.14}$ | 22.60 $_{\pm 0.08}$ |
| H-DPO ($\alpha = 0.95$) | **5.50** $_{\pm 0.80}$ | 0.73 $_{\pm 0.52}$ | **14.27** $_{\pm 0.19}$ | 22.93 $_{\pm 0.27}$ |
| H-DPO ($\alpha = 0.9$) | 4.40 $_{\pm 0.19}$ | 0.57 $_{\pm 0.28}$ | 14.13 $_{\pm 0.12}$ | **23.67** $_{\pm 0.10}$ |

## A.4  Evaluation of Diversity

For the evaluation of diversity in Table 2, we used entropy, Self-BLEU (Zhu et al., 2018), and Distinct-1, -2 (Li et al., 2016). Regarding the measurement of entropy, we used 200 prompts from the UltraFeedback (Cui et al., 2023) test dataset, which was used in the training of DPO, and generated 25 responses for each prompt. The maximum length of the responses was limited to 512, and the entropy was calculated using the log probability of each response, normalized by the response length. Self-BLEU and Distinct-1, -2 were also calculated using the same responses based on Zhu et al. (2018) and Li et al. (2016).

## A.5  Other Details

Table 4: Average scores of DPO and H-DPO on various tasks.

| | GSM8K↑ | HumanEval↑ | MMLU-Pro↑ | IFEval↑ |
|---|---|---|---|---|
| DPO ($\alpha = 1, \beta = 0.01$) | 26.40 $_{\pm 1.76}$ | 28.77 $_{\pm 0.45}$ | 31.83 $_{\pm 0.17}$ | 59.63 $_{\pm 0.72}$ |
| DPO ($\alpha = 1, \beta = 0.009$) | 25.13 $_{\pm 1.22}$ | 26.37 $_{\pm 1.34}$ | 31.93 $_{\pm 0.10}$ | 59.53 $_{\pm 0.35}$ |
| H-DPO ($\alpha = 0.9, \beta = 0.01$) | **28.83** $_{\pm 2.32}$ | **29.63** $_{\pm 0.45}$ | **32.30** $_{\pm 0.17}$ | **60.93** $_{\pm 0.50}$ |

MMLU-Pro was evaluated using the official implementation from Wang et al. (2024d). IFEval and GSM8K were implemented using Gao et al. (2024), where IFEval was evaluated in a 0-shot setting, and GSM8K was evaluated in an 8-shot setting. HumanEval was evaluated using the official implementation from Chen et al. (2021). MMLU-Pro and IFEval were evaluated using one sampling for all test data, and the average accuracy and standard error at a temperature of 0 are shown in Table 1 and 3. For GSM8K, 200 test data were used, and for HumanEval, all test data were used, generating 200 responses for each prompt to calculate pass@$k$ based on Chen et al. (2021), as shown in Figure 3 and 4. The results for pass@1 at a temperature of 0.1 are shown in Table 1 and 3. The results for varying temperatures in MMLU-Pro, GSM8K, HumanEval, and IFEval are shown in Figure 2 to 4 and 10.

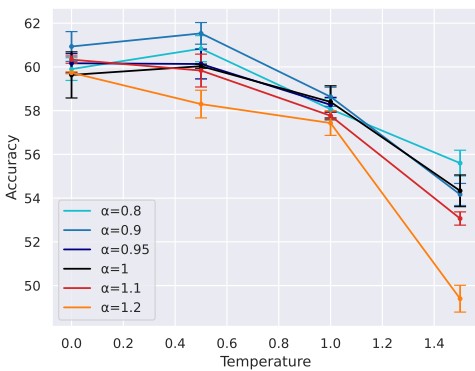

Figure 10: Accuracy on IFEval with various temperatures.

## B Broader Impact

As this paper primarily focuses on the algorithmic contributions to fine-tuning language models using DPO, its direct societal impact is limited. However, the application of our methodology, particularly in the context of RLHF, requires careful consideration of the feedback process. The individuals providing feedback play a crucial role in shaping the behavior of the language model. Ensuring that the feedback discourages harmful, malicious, or unethical outputs is essential for aligning the model with societal norms and ethical standards.

