# OpenReview forum: "Entropy Controllable Direct Preference Optimization"
_TMLR — Rejected by TMLR_

### Review · Reviewer_xBtv · 2024-12-04

**Summary Of Contributions:**

This paper proposes H-DPO by adding an entropy control term to DPO's objective function. The key insight is that even reverse KL divergence may fail to achieve mode-seeking fitting in certain cases. The authors introduce a hyperparameter α to control the entropy of policy distribution, theoretically analyze its effects through Gaussian mixture models, and validate the method on various tasks using Mistral-7B.

**Audience:**

Yes

**Claims And Evidence:**

Yes

**Requested Changes:**

1. Broaden model evaluation beyond Mistral-7B, testing H-DPO with at least one smaller model (e.g., 7B) and one larger model (e.g., 70B) to demonstrate the method's generalizability across model scales.

2. Include a wider range of tasks, particularly:
   - Open-ended generation tasks to demonstrate performance on less constrained problems
   - Dialogue tasks since they are a major application of DPO
   - Tasks with more diverse performance metrics beyond accuracy and pass@k

3. Add a detailed analysis section investigating when and why the method performs inconsistently on HumanEval, comparing failure cases with successful ones.

4. Provide more concrete guidance for selecting α values, possibly through an analytical relationship between task characteristics and optimal α.

**Strengths And Weaknesses:**

Strengths

1. The paper identifies an interesting failure case of DPO where reverse KL divergence loses its mode-seeking property, supported by theoretical analysis and visualization.
2. The proposed solution H-DPO is elegant and extremely simple to implement, requiring only a coefficient change in the loss function.
3. The empirical results show consistent improvements on GSM8K and MMLU-Pro tasks with different α values, demonstrating the effectiveness of entropy control.

Weaknesses

1. The evaluation is limited to Mistral-7B model only, making it hard to conclude whether the improvements generalize across model architectures and sizes.
2. The performance gains on HumanEval are unstable and not significant in serveral pass@k metrics, suggesting the method may not be universally effective.
3. The paper lacks sufficient task diversity, mostly focusing on mathematical and coding tasks while missing important domains like open-ended generation and dialogue.

---

> ### Author Response · Authors · 2024-12-30
> **Response**
>
> Thank you for the valuable comments!
>
> **Evaluation on Different Models and Tasks**
>
>
> Regarding experiments with different models, due to limited computational resources, it is challenging for us to conduct experiments on larger models, such as 70B. As noted in related works on DPO [1, 2], only one model, such as a 2.8B or 7B, is typically employed.
> As for smaller models, we have added the results for the Llama-3.2-1B model in Table 3. Given the difficulty of the tasks and the inherently low performance of the base model, consistent improvements were not observed in HumanEval. However, we did observe performance improvements in other tasks.
> For evaluation on open-ended tasks, we attempted scoring-based MT-bench. However, due to large variance in the scores and the current number of seeds, we could not obtain consistent results regarding changes in $\alpha$. As a result, our paper does not claim improvements in dialogue tasks.
>
> **Analysis of Results and Alpha Selection**
>
> We have added an analysis of the HumanEval results and the selection of the hyperparameter $\alpha$ in Section 5.4.
>
>
> [1] Rafailov, R., Sharma, A., Mitchell, E., Ermon, S., Manning, C. D., & Finn, C. (2023). Direct preference optimization: Your language model is secretly a reward model. Proceedings of the 37th International Conference on Neural Information Processing Systems (NeurIPS 2023).
>
> [2] Song, F., Yu, B., Li, M., Yu, H., Huang, F., Li, Y., & Wang, H. (2024). Preference ranking optimization for human alignment. Proceedings of the AAAI Conference on Artificial Intelligence, 38(17), 18990-18998.

---

### Review · Reviewer_D95U · 2024-12-14

**Summary Of Contributions:**

This paper provides the following insights: (1) The reverse KL divergence regularizer used in DPO objectives is not optimal because it can cause the mode of the fitted distribution to fail to capture the mode of the reference distribution, i.e., poor mode-seeking fitting. To highlight this point, the authors show a toy example using a Gaussian mixture. (2) To improve mode-seeking capabilities, the authors proposed a more generalized version of DPO, with an additional hyperparameter controlling the entropy. This is empirically justified through evaluations covering different datasets/tasks (GSM8K, HumanEval, MMLU-Pro, IFEval).

**Audience:**

Yes

**Claims And Evidence:**

No

**Requested Changes:**

Please provide more rigorous experimentation on either more complex setups or distributions, or provide theoretical insights to justify the following claims:

1) Reverse KL divergence may not necessarily result in mode-seeking behavior when fitting the target distribution.

2) H-DPO, by controlling entropy, guarantees an improvement in mode-seeking behavior.

**Strengths And Weaknesses:**

> Strengths

1. The paper is well-written in general and the problem statement has been well-articulated throughout the paper.

2. The proposed approach shows empirical improvements in performance for different tasks.

> Weaknesses

1. The insight that optimizing reverse KL divergence may not result in a mode-seeking fitting with the target distribution is interesting. However, the current version of the paper does not sufficiently justify this claim. The authors attempt to motivate this problem using a simple toy example, but more rigorous experimentation is needed to validate the claims.

For example, the following sentence in Section 4.1, “In this case, if $\pi$ is a language model, it is likely to generate from valleys where $\pi_{\text{ref}}$ has a low probability, possibly leading to degraded performance of $\pi$.”, has been claimed solely based on one toy example.

2. Additionally, for complex reference distributions, how does H-DPO guarantee that minimizing $\alpha$ would make $\pi$  fit one of the modes of $\pi_{\text{ref}}$? I think providing some theoretical insights to justify the claim would make the paper stronger.

---

> ### Author Response · Authors · 2024-12-28
> **Response**
>
> Thank you for your valuable feedback!
>
>
> As per your request, we have added experiments using more complex reference distributions in Figures 5, 7, and 8. In these figures, we employed Gaussian mixture models with 2, 3, and 4 Gaussian components as the reference distributions. Each row in the figures represents results where the mean spacing between the Gaussian components is varied.
>
> In many cases, minimizing the KL divergence (with $\alpha = 1$ ) fails to achieve mode-seeking fitting. However, by reducing $\alpha$ , mode-seeking fitting that focuses on the mode with the highest probability becomes feasible across all cases.
>
> The insight here is that mode-seeking fitting needs to reduce variance compared to the reference distribution by discarding unfitted modes. However, minimizing the reverse KL divergence leads to a larger reverse KL value when the variance deviates from the reference distribution. In contrast, $D_\alpha (p || q)$ is defined as:
>
> $ D_\alpha (p || q) = D_\text{KL}(p || q) + (1 - \alpha) H(p) $
>
> which includes an additional entropy term. Minimizing $D_\alpha$ with $\alpha > 1$ reduces the variance, making it well-suited for mode-seeking fitting.

---

### Review · Reviewer_Bear · 2024-12-20

**Summary Of Contributions:**

This paper introduces a modification to Direct Preference Optimization for post-training large language models (LLMs). The authors claim that minimizing reverse KL divergence in DPO can fail to capture a mode of the reference distribution, potentially degrading the policy’s performance. To address this, they propose to rewrite the loss function to directly control the entropy, claiming that this enhances mode-seeking behavior and improves the sharpness of the resulting distribution.

**Audience:**

Yes

**Broader Impact Concerns:**

It might be worth discussing who is going to provide the human feedback, and how.

**Claims And Evidence:**

No

**Requested Changes:**

Presenting compelling evidence addressing Weakness 1 is an absolute must for me to even consider acceptance.

**Strengths And Weaknesses:**

**Strengths**
It is a timely topic that is certainly interesting to TMLR's audience.

**Weaknesses**

1. **Fundamental Flaws in Theoretical Basis**. The authors present a toy example where they fit a unimodal Gaussian to a bimodal Gaussian mixture. They claim that minimizing the reverse KL divergence (as in standard variational inference) fails to capture one of the modes - in direct contradiction to established theory. To me, the only certain conclusion you may draw from this experiment is that the optimization has failed. Instead, the authors put forth this (questionable) observation as the basis for their proposed modification, casting doubt on the entire approach.

2. **Weak Motivation for Mode-Seeking Behavior**. The reasoning behind the need for mode-seeking behavior is unclear and unconvincing. Stronger theoretical or empirical justification is required to make this case compelling.

3. **Writing Quality**. The paper’s writing is difficult to follow in places and would benefit significantly from improved clarity and organization.

4. **Unclear Improvement over Prior Work**. I struggle to see any meaningful advancement over Wang et al. (2024), which already provides a robust study of f-divergences, addressing similar concerns.

**Unreviewed Experimental Section**. Due to my belief that the theoretical foundation of the paper is fundamentally flawed, I could not justify a detailed review of the experiments.

---

> ### Author Response · Authors · 2024-12-28
> **Response**
>
> Thank you for your comments! We provide our responses below.
>
> **Fundamental Flaws in Theoretical Basis**
>
> Regarding the comment, “To me, the only certain conclusion you may draw from this experiment is that the optimization has failed,” we do not think that our results, particularly the left side of Figure 1, is due to an optimization failure. In the experiment, we divide the plausible ranges of the mean and standard deviation of the Gaussian distribution to be estimated into 100x100 grids and search the grid that minimizes the divergence, which is Monte-Carlo approximated with 1e6 samples each. In fact, the values of $D_{KL}(\pi || \pi_\text{ref})$ in the left side and right side of Figure 1 are 0.336 and 0.721, respectively, indicating that the mode-covering $\pi$ achieves a smaller $D_\text{KL}$.
>
> The top right of Figure 6 illustrates the $D_{KL}(\pi || \pi_\text{ref})$ for various location and scale parameters of the normal distribution $\pi$. This visualization clearly demonstrates that $D_\text{KL}$ is smallest when the fitting is mode-covering in this setting, indicating that it is not an optimization failure.
>
> While it is true that minimizing the reverse KL divergence tends to lead to mode-seeking estimation compared to other f-divergences, it does not always succeed, and failure cases exist. To further support this, we added experiments using more complex $\pi_\text{ref}$ in Figures 5, 7, and 8, and observed similar results in these cases as well. We have included the code used for generating the plots in the supplementary material.
>
> To our knowledge, there is no theoretical result that precludes the possibility of a unimodal Gaussian minimizing the KL divergence against a Gaussian mixture while located between the mixture components as in Figure 1. We would appreciate it if you could point out theoretical results of such.
>
> **Unclear Improvement over Prior Work**
>
> Wang et al. (2024) demonstrated that reverse KL divergence outperforms other f-divergences in terms of accuracy. We attribute this to the fact that reverse KL is better suited for mode-seeking estimation compared to other f-divergences like forward KL. Based on this, we proposed a regularization to enhance the mode-seeking property of reverse KL divergence. This regularization is not included in traditional f-divergences and was not explored in Wang et al. (2024).
>
> We have also added a broader impact section. Please let us know if there are any other concerns.

---

> > ### Comment · Reviewer_Bear · 2025-01-08
> > **Empirical observation is OK**
> >
> > Thanks for providing the code and additional experiments related to Figure 1 - I stand corrected. My (faulty) intuition was based on the argument that VI is zero-forcing, as discussed e.g. in section 10.1.2 of Bishop's classical book from 2006. However, this logic is not entirely applicable in your example, since the overlap between the mixture components is substantial.
> >
> > That said, my other concerns largely remain.

---

> > > ### Author Response · Authors · 2025-01-09
> > > **Response by Authors**
> > >
> > > Thank you very much for your response! Below, we provide our replies to the other concerns.
> > >
> > > **Weak Motivation for Mode-Seeking Behavior**
> > >
> > > When mode-seeking fitting fails, as illustrated in the left panel of Figure 1, samples are more likely to be drawn from the low-probability regions in the valleys of $\pi_\text{ref}$. Avoiding this issue serves as the motivation for aiming at mode-seeking estimation. Furthermore, Wang et al. (2024) demonstrated that reverse KL regularization outperforms forward KL regularization, providing empirical justification that the mode-seeking property is crucial.
> > >
> > > **Writing Quality**
> > >
> > > Thank you for your feedback. We would greatly appreciate it if you could point out specific parts that you found difficult to follow.
> > >
> > > If you have any further concerns, please feel free to let us know.

---

### Decision · Action_Editor_g8Gp · 2025-02-19

**Recommendation:** Reject

**Comment:**

This paper is not ready for publication yet. The experiments and the writing seem to be rushed. The proposed approach is straightforward and incremental in contributions, introducing only one more hyperparameter gamma in front of the log ratios of pi_theta over rejected and accepted responses. This is a minor modification to the original DPO formulation. However, this is not a reason for rejecting this paper, if the results were sufficiently supported by experimental and theoretical data, it could still have been a meaningful contribution. However, as mentioned in the Claims and Evidence section, as well as by the reviewers, the claims in this paper are not supported very well by the evidential data.

I recommend that the authors work on the writing of the paper, for example, by providing a sufficient explanation of how Figure 1 was obtained and explaining more about why reverse-KL is mode-covering in that example. Moreover, looking into gathering more experimental and theoretical data to justify their claims, as well as experiments on a broader range of models and datasets to support the claims that H-DPO consistently improves over DPO for the said reasons, and then consider resubmitting again.

**Audience:**

The audience would be the researchers working on alignment and post-training of LLMs.

**Claims And Evidence:**

## Summary

This paper proposes an extension to the DPO algorithm by adding an entropy regularization. The paper claims that the reverse-KL regularization will not have a mode-seeking behavior if there is a substantial overlap between different modes of pi_ref when optimizing the min KL(pi_\theta|pi_ref). To achieve the mode-seeking behavior, the authors introduce the H-DPO approach. They claim their method improves over DPO when evaluated over the mathematical reasoning tasks.

---

## Claims

**Claims on mode-seeking:** Overall, I think the claims made by the paper concerning the mode-covering behavior of reverse-KL in alignment settings and why we need mode-seeking behavior instead of mode-covering are not very well justified empirically and experimentally.

**Experimental results are lacking.** As noted by the reviewers, this paper fails to provide enough experimental results to prove the efficacy of their proposed approach. Their evaluation is limited to the Mistral-7B model only, making it hard to conclude whether the improvements generalize across model architectures and sizes. No scaling study has been conducted to investigate the applicability of the proposed approach to the larger models. The performance gains on HumanEval are unstable and insignificant in several pass@k metrics, suggesting the method may not be universally effective. The paper lacks sufficient task diversity, mainly focusing on mathematical and coding tasks while missing important domains like open-ended generation and dialogue.

**Resubmission Of Major Revision:**

The authors may consider submitting a major revision at a later time.